# Identification of a Novel Hepacivirus in Southeast Asian Shrew (*Crocidura fuliginosa*) from Yunnan Province, China

**DOI:** 10.3390/pathogens12121400

**Published:** 2023-11-28

**Authors:** Ling Guo, Bei Li, Peiyu Han, Na Dong, Yan Zhu, Fuli Li, Haorui Si, Zhengli Shi, Bo Wang, Xinglou Yang, Yunzhi Zhang

**Affiliations:** 1Yunnan Province Key Laboratory of Anti-Pathogenic Plant Resources Screening, Yunnan Province Key University Laboratory of Zoonoses Cross-Border Prevention and Quarantine, Institute of Preventive Medicine, School of Public Health, Dali University, Dali 671000, China; g183130@163.com (L.G.); hanpeiyu1511@gmail.com (P.H.); 15133636523@163.com (N.D.); d258528@126.com (F.L.); 2Chongqing Jiangbei District Center for Disease Control and Prevention, Chongqing 400020, China; 3Key Laboratory of Special Pathogens and Biosafety, Wuhan Institute of Virology, Chinese Academy of Sciences, Wuhan 430071, China; libei@wh.iov.cn (B.L.); zhuyan@wh.iov.cn (Y.Z.); sihaorui18@mails.ucas.ac.cn (H.S.); zlshi@wh.iov.cn (Z.S.); 4Department of Biomedical Sciences and Pathobiology, Virginia-Maryland College of Veterinary Medicine, Virginia Polytechnic Institute and State University, Blacksburg, VA 24061, USA; bowang@vt.edu; 5Yunnan Key Laboratory of Biodiversity Information, Kunming Institute of Zoology, Chinese Academy of Sciences, Kunming 650023, China

**Keywords:** small mammal, Southeast Asian shrew, *Crocidura fuliginosa*, *Flaviviridae*, *Hepacivirus*

## Abstract

The genus *Hepacivirus* contains single-stranded positive-sense RNA viruses belonging to the family *Flaviviridae*, which comprises 14 species. These 14 hepaciviruses have been found in different mammals, such as primates, dogs, bats, and rodents. To date, *Hepacivirus* has not been reported in the shrew genus of *Crocidura*. To study the prevalence and genetic evolution of *Hepacivirus* in small mammals in Yunnan Province, China, molecular detection of *Hepacivirus* in small mammals from Yunnan Province during 2016 and 2017 was performed using reverse-transcription polymerase chain reaction (RT-PCR). Our results showed that the overall infection rate of *Hepacivirus* in small mammals was 0.12% (2/1602), and the host animal was the Southeast Asian shrew (*Crocidura fuliginosa*) (12.5%, 2/16). Quantitative real-time PCR showed that *Hepacivirus* had the highest viral RNA copy number in the liver. Phylogenetic analysis revealed that the hepaciviruses obtained in this study does not belong to any designated species of hepaciviruses and forms an independent clade. To conclude, a novel hepacivirus was identified for the first time in *C. fuliginosa* specimens from Yunnan Province, China. This study expands the host range and viral diversity of hepaciviruses.

## 1. Introduction

The family *Flaviviridae* currently contains four genera, including *Orthoflavivirus, Pestivirus*, *Hepacivirus*, and *Pegivirus* (https://ictv.global/report/chapter/flaviviridae/flaviviridae, accessed on 15 November 2023). *Hepavirius* is a genus of a group of genetically diverse viruses with small spherical envelops and positive-sense RNA genomes of approximately 8.9–10.5 kb [1], which contain a single long open reading frame (ORF) flanked by 5′- and 3′-terminal untranslated regions (UTRs). The ORF encodes a polyprotein of approximately 3000 amino acids (aa), which is further proteolytically cleaved by host and viral proteases into structural proteins (Core, E1, and E2) and nonstructural proteins (p7, NS2, NS3, NS4A, NS4B, NS5A, and NS5B) [2].

*Hepacivirus* comprises 14 species (*Hepacivirus bovis*, *colobi*, *equi*, *glareoli*, *hominis*, *macronycteridis*, *myodae*, *norvegici*, *otomopis*, *peromysci*, *platyrrhini*, *ratti*, *rhabdomysis*, *vittatae*) according to the International Committee on Taxonomy of Viruses (ICTV) classification (https://ictv.global/taxonomy, accessed on 15 November 2023). Members of specific hepacivirus species show characteristically distinct host ranges, including horses and possibly dogs (*Hepacivirus A*, renamed *Hepacivirus equi*), tamarins and potentially other New World primates (*Hepacivirus B*, renamed *Hepacivirus platyrrhini*), humans (*Hepacivirus C*, renamed *Hepacivirus hominis*), colobus monkeys (*Hepacivirus D*, renamed *Hepacivirus colobi*), various species of rodents (*Hepacivirus E*, *F*, *G*, *H*, *I* and *J*, renamed *Hepacivirus peromysci*, *myodae*, *ratti, norvegici*, *rhabdomysis*, *glareoli*, respectively), bats (*Hepacivirus K*, *L*, and *M*, renamed *Hepacivirus macronycteridis*, *vittatae*, and *otomopis,* respectively), and cows (*Hepacivirus N*, renamed *Hepacivirus bovis*) [2,3,4,5] (https://ictv.global/report/chapter/flaviviridae/flaviviridae/hepacivirus, accessed on 15 November 2023). However, apart from the human hepatitis C virus (HCV, *Hepacivirus hominis*), the molecular biology of the other above-mentioned hepaciviruses has not been well characterized.

Recently, a novel hepacivirus in three-toed sloths (*Bradypus variegatus*) from Costa Rica was detected and could represent a novel hepacivirus species [6]. More importantly, a study suggested that host switches of hepacivirus were possible, such as from rodents to sloths, and rodents have played a prominent role in *Hepacivirus* genealogy [6]. Although small rodents and bovines are recognized as important reservoirs of hepaciviruses, few investigations have addressed hepacivirus in other small wild mammals in China [7,8,9,10,11,12,13]. This study aimed to close this knowledge gap, and hepacivirus screening was conducted on a total of 1602 small wild mammals captured in a wide geographical area, Yunnan Province, in China from 2016 to 2017.

## 2. Materials and Methods

### 2.1. Samples Collection and Ethical Statements

From 2016 to 2017, small mammal samples were collected at 12 sites in Yunnan Province, China (Figure 1 and Table 1). The sample trapping was performed using cage-type traps using freshly fried fritters as bait. The collected small mammals were brought back to the laboratory and were exsanguinated during isoflurane anesthesia on a warming blanket to minimize distress. Species were initially identified based on morphology, followed by further molecular identification of the species via sequence analysis of the mitochondrial (mt)-cytochrome b (Cytb) gene [14]. The samples were dissected in an aseptic environment of a biosafety level 2 (BSL-2) laboratory, and the heart, liver, spleen, lung, kidney, and intestine tissues were collected in 2 mL cryogenic vials (CORNING, Shanghai, China) and stored temporarily in liquid nitrogen. The samples were then stored at −80 °C before further laboratory analyses. This study was approved by and conducted according to the guidelines of the Animal Ethics Committee of Dali University (approval no. DLDXLL2017007).

### 2.2. Viral RNA Extraction and RT-PCR for Hepacivirus

Under aseptic conditions, approximately 0.1 g of liver tissue sample was cut into GeneReady Animal PIII crushing tubes (Life Real, Hangzhou, China) and diluted with 1 mL of sterilized phosphate-buffered saline (PBS) to prepare a 10% weight per volume (% *w*/*v*) tissue suspension, followed by grinding in a GeneReady Ultimate grinder (Life Real, Hangzhou, China). The tissue suspension was then clarified via centrifugation at 10,000× *g* for 15 min and filtered through a 0.45 mm membrane. Viral RNA was extracted from liver tissue using the Viral RNA mini kit (Qiagen, Hilden, Germany) following the manufacturer’s protocol. Hepacivirus RNA was detected using a 400 bp segment of NS3 analysis via reverse-transcription-nested polymerase chain reaction (RT-PCR) with degenerate primers (Hepaci-F1: GCICCIACIGGIAGYGGIAA; Hepaci-R1: CCIGTCATIAGRGCRTCIGT; Hepaci-R2: CARTCIRTIACIGARTCRAARTYICC; Hepaci-F2: TAYGAYGTIATIATITGYGAYGARTG; Hepaci-F3: TAYGAYGTIATIATITGYGAYGA; Hepaci-F4: GCIACIGCIACICCICCIGG; Hepaci-R3: CCIGTCATIGRGCRTCIGT; Hepaci-R4: CARTCIRTIACIGARTCRAARTYICC), as described previously [8,15,16]. PCR was performed using Super ScriptⅢ One-Step RT-PCR System with Platinum Taq DNA Polymerase (Thermo Fisher Scientific, Waltham, MA, USA), following the manufacturer’s instructions. First-round RT-PCR was performed with 5 µL of RNA, 400 nM each of 1st-round primers or an equimolar mix of primers, 1 µg of bovine serum albumin, 0.2 mM of each dNTP, and 2.4 mM of MgSO_4_. Second-round 50 µL Platinum Taq reactions involved the use of 1 µL of 1st-round PCR product, 2.5 mM MgCl_2_, and 400 nM each of the 2nd-round primers. First-round RT-PCR reactions used a touchdown protocol with reverse-transcription at 48 °C for 30 min, denaturation at 95 °C for 3 min, followed by PCR 10 cycles of 15 s at 94 °C, 20 s at 60 °C, with a decrease of 1 °C per cycle, and extension at 72 °C for 45 s, followed by another 40 cycles at 50 °C annealing temperature. Second-round reactions used the same cycling protocol without the RT step. The PCR products were examined on 1.2% agarose gel for electrophoretic analysis and observed with a UV gel imager for the presence of target bands with a specific molecular weight (about 400 bp). The objective PCR bands underwent gel purification with a MinElute Gel Extraction Kit (Qiagen, Hilden, Germany), and they were sequenced with both forward and reverse primers by Tsingke Biotechnology (Wuhan, China).

### 2.3. Viral Complete Genome Sequencing and Assembly

The entire genome sequence of the two hepacivirus RNA-positive samples was evaluated using next-generation sequencing using an Illumina HiSeq 3000 system (Illumina, San Diego, CA, USA). The sequencing reads were assembled and analyzed using MEGAHIT (v1.2.9) and Geneious (v10.0.9) algorithms, respectively. The remaining genome gaps were filled using specific primers through conventional PCR. The genome end sequences were obtained via rapid amplification of cDNA ends (RACE) using a Clontech SMARTer RACE 5′ and 3′ Kit (Takara, Kusatsu, Japan) and via poly (A) tailing [17].

### 2.4. Phylogenetic Analysis

The sequence alignments contained the genomic nucleotide sequences of the virus identified in this study together with 27 sequences obtained from representative members of the genus *Hepacivirus* using ClustalW. Phylogenetic trees were inferred for coding sequences of the polyprotein genes using the maximum likelihood (ML) method and the Jukes–Cantor model implemented in MEGA X software. To assess support for individual nodes, 1000 bootstrap replicates were obtained following the same procedure. The phylogenetic trees were visualized using Adobe Illustrator CC 2018.

RDP4 was used to detect potential recombination sequences and parental sequences in gene fragments and entire ORF sequences. Potential recombination events were considered significant when the *p*-values of more than 6 detection methods were all <0.05. SimPlot software 3.5.1 was used for further verification.

### 2.5. Quantitative Real-Time RT-PCR (qRT-PCR)

Using a HiScript II One Step qRT-PCR SYBR Green kit (Vazyme Biotech, Nanjing, China), quantitative RT-PCR was performed according to the manufacturer’s instructions. A positive control from the partial genome RNA transcript of the DNA template was applied using a MAXI script Kit (Applied Biosystems, Waltham, MA, USA) in vitro. Specific primers (5′–TCGGTGATGTCTTTATGAGGC–3′ and 5′–CCTTGGTCTGACTTTGATCTTAAC–3′) were designed based on the sequence obtained in this study. PCR conditions were as follows: 50 °C for 15 min; 95 °C for 5 min; 40 cycles at 95 ˚C for 10 s, 60 °C for 30 s, 95 °C for 15 s, and a final extension at 60 °C for 1 min.

## 3. Results

### 3.1. Hepacivirus Detection

Overall, 1602 small mammals—comprising 21 species, 11 genera, 4 families, and 3 orders—were trapped in 12 counties of Yunnan Province (Figure 1 and Table 1). Two liver tissue samples (0.12%, 2/1602) of *Crocidura fuliginosa* (commonly known as Southeast Asian shrew) were positive for hepacivirus nucleic acids. Taxonomically, the Southeast Asian shrew is a species of insectivore in the genus *Crocidura*, family *Soricidae*, order Eulipotyphla, which is widely spread throughout Southeastern Asian countries (https://en.wikipedia.org/wiki/Southeast_Asian_shrew, accessed on 15 November 2023).

One *Crocidura fuliginosa* specimen was collected from Nanjian County (termed Hepacivirus Cf/NJ12), and another was collected from Xiangyun County (termed Hepacivirus Cf/XY174) in 2017. The positive rate of hepacivirus infection among *Crocidura fuliginosas* was 12.5% (2/16). Specifically, Hepacivirus Cf/NJ12 had 73.7% nucleotide identity with Hepacivirus Cf/XY174 based on the partial NS3 sequence. In particular, Hepacivirus Cf/NJ12 shares a high similarity (69.25%) with the Asian house shrew (*Suncus murinus*) hepacivirus SZCDC19 (GenBank no. MF775363) based on the partial NS3 sequence. In contrast, Hepacivirus Cf/XY174 is most similar (68.90%) to *Suncus* sp. hepacivirus SoSm-HepaV/Cs2009 (GenBank no. MT085224) based on the partial NS3 sequences.

### 3.2. Genome Characterization

Because of the poor sequence assembly quality of Hepacivirus Cf/XY174, only the complete genomic sequence of Hepacivirus Cf/NJ12 was determined, which was 10,383 nucleotides in length and with a G + C content of 54.5% (genome warehouse number: GWHBHEC01000000). Similar to other known hepaciviruses, the genome of Hepacivirus Cf/NJ12 containseda single ORF encoding a polyprotein of 3167 amino acids flanked by the 5′- and 3′-untranslated regions (UTR) (822 and 57 nucleotides, respectively), and the polyprotein was putatively cleaved in ten proteins (Core, E1, E2, P7, N2, N3, N4A, N4B, N5A, N5B) (Figure 2). The predicted polyprotein shared the highest identity of 65.5% (nucleotide) and 72.1% (amino acid) with a hepacivirus detected in a striped field mouse (Species: *Apodemus agrarius*) (GenBank no. QUS47367) in 2016 from Wenzhou City, China. However, the Hepacivirus Cf/NJ12 had only an identity of 40.6% (nucleotide) and 26.6% (amino acid) with the human HCV reference strain H77 (GenBank no. AF009606). Comparative sequence analysis showed that the Hepacivirus Cf/NJ12 ORF detected had 30.3–37.7% nucleotide and 20.6–31.1% amino acid identities with other known hepacivirus species (Table 2).

### 3.3. Phylogenetic and p-Distance

Phylogenetic analysis based on the nucleotide sequence of the ORF region showed that the hepacivirus strain detected in *Crocidura fuliginosa* formed an independent clade (Figure 3A). Notably, the novel Hepacivirus Cf-NJ12 is clustered with other four hepaciviruses previously found in different rodents, indicating that it branched out from their ancestor long before the other four rodent hepaciviruses. However, it should also be noted that the rodent hepaciviruses and bat hepaciviruses dispersed in the phylogenetic tree of the genus *Hepacivirus*, the precise relationship between shrew hepaciviruses and rodent hepaciviruses is yet to be determined and awaits the determination of further complete genomes of genetically heterogeneous hepaciviruses in other small mammals.

In addition to the Southeast Asian shrew (*Crocidura fuliginosa*) (Figure 3B), the genus *Crocidura* actually contains three additional species, namely, the Asian gray shrew (*Crocidura attenuata*), Horsfield’s shrew (*Crocidura horsfieldii*), and Voracious shrew (*Crocidura vorax*), which presumably harbor their own hepaciviruses. It would be appealing to elucidate the relationship of hepaciviruses in *Crocidura* shrews.

We did not find any evidence of recombination in the Hepacivirus Cf/NJ12 genome. The amino acid p-distances of the conserved regions 996–1437 and 2693–3115 (relevant to positions 1123–1566 and 2536–2959 of the HCV genotype 1a of H77 reference strain (AF011751) of Hepacivirus Cf/NJ12 range within 0.518–0.590 and 0.464–0.546 with known species of the genus *Hepacivirus*, respectively (Table 2). The ICTV species demarcation criteria for this genus state that a novel hepacivirus must exhibit a p-distance greater than 0.25 between amino acids 1123 and 1566 of NS3 and greater than 0.3 between amino acids 2536 and 2959 NS5, as numbered in the HCV genotype 1a of H77 reference strain (AF011751) [3]. Therefore, we propose that Hepacivirus Cf/NJ12 should constitute a new hepacivirus species.

### 3.4. Quantification of Hepacivirus in Different Tissues Using qRT-PCR

To determine the viral loads of hepacivirus in different organs of the hepacivirus-RNA-positive specimen, viral genomic RNA copies were quantified in several tissue samples, including the liver, spleen, kidney, and intestine of Hepacivirus Cf-NJ12 via qRT-PCR. The results showed that hepacivirus RNA was detected in all tested tissues, and the viral load in *Crocidura fuliginosa* ranged from 5 × 10^4^ to 3.14 × 10^5^ copies/gram of tissue. Moreover, the viral load in the liver was numerically higher than in other tissues and reached 3.14 × 10^5^ copies/g tissues (Figure 4). Whether shrew hepacivirus causes persistent or chronic infection in its host as human HCV remains unclear.

## 4. Discussion

So far, the genus *Hepacivirus* includes 14 species with a wide range of host reservoirs, including humans, Old World colobus monkeys, various rodent and bat species, European and African cattle, horses, and donkeys (https://ictv.global/report/chapter/flaviviridae/flaviviridae/hepacivirus, accessed on 15 November 2023). Rodents are the natural reservoirs of various hepaciviruses, and they may be the major hosts driving the spread of hepaciviruses between different host orders [6]; among the rodential hosts of six hepaciviruses (originally named *Hepacivirus E* to *J*) in *Hepacivirus* according to ICTV, *Hepacivirus peromysci* (*E*) was detected in deer mouse (*Peromysus maniculateus*) [18], *Hepacivirus myodae* (*F*) and *Hepacivirus glareoli* (*J*) were detected in bank vole (*Myodes glareolus*) [15], *Hepacivirus ratti* (*G*) and *Hepacivirus norvegici* (*H*) were detected in Norway rat (*Rattus norvegicus*) [19], and *Hepacivirus rhabdomysis* (*I*) was detected in South African four-striped mouse (*Rhabdomys pumilio*) [15]. In a previous study, among 4770 rodent sera and organ specimens (41 species) from around the world, three highly divergent rodent hepacivirus clades were detected in 27 (1.8%) of 1465 European bank voles (*Myodes glareolus*) and 10 (1.9%) of 518 South African four-striped mice (*Rhabdomys pumilio*) [15]. However, in this survey, hepacivirus was only detected in one of three species of shrews (order Eulipotyphla) with a detection rate of 12.5% (2/16), but none was detected in 18 species of rodents (order Rodentia); whether other animal species could harbor hepacivirus remains to be determined. Moreover, which small mammals have higher hepacivirus genetic diversity and detection rates is largely unknown.

Southeast Asian shrew (*Crocidura fuliginosa*) is widely distributed in Southeast Asia (https://www.iucnredlist.org/species/40631/115176525, accessed on 15 November 2023), although the current population trend is unknown. Among the total of 16 Southeast Asian shrew samples collected from seven counties, a novel hepacivirus was detected in one sample from Xiangyun County and one of three samples from Nanjian County. A plausible explanation for the results is that Xiangyun County and Nanjian County are connected (Figure 1) and belong to the Wuliang Mountains, which are known for their abundant biodiversity under suitable geographic conditions, including temperature, humidity, altitude, longitude, and latitude. Additionally, the ecosystems of the two counties are similar, so are altogether suitable for the growth and reproduction of small mammals such as the Southeast Asian shrew. In the present study, the novel hepacivirus was only found in Xiangyun County and Nanjian County, reflecting that the distribution of these viruses may be limited to these two regions. However, it still cannot be ruled out that with the increase in the sample size, the virus would also be detected in other locations. Nonetheless, whether Southeast Asian shrews in other counties of Yunnan Province harbor this novel hepacivirus remains to be determined. Moreover, considering the movement of shrews due to ecological changes, we cannot exclude the possibility of hepacivirus spread in the future. Since there is thus far no evidence of zoonotic infection of hepacivirus, it is unlikely that the novel shrew hepacivirus can cross species barriers to infect humans.

In addition to the hepaciviruses that have already been reported by ICTV, novel hepaciviruses have been gradually discovered around the world. For example, novel hepaciviruses were found in Asian house shrew (*Suncus murinus*) in Shenzhen, China [8]; montane grass mouse (*Akodon montensis*), delicate vesper mouse (*Calomys tener*), black-footed pigmy rice rat (*Oligoryzomys nigripes*), hairy-tailed bolo mouse (*Necromys lasiurus*) and house mouse *(Mus musculus*) from São Paulo State, Brazil [5]; long-tailed ground squirrel (*Spermophilus undulates*) from Xinjiang Uygur Autonomous Region, China [9]; cows from Yunnan, China [11]; common brushtail possum (*Trichosurus Vulpecula*) from Sydney, Australia [20]; and vrown-throated three-toed sloth (*Bradypus variegatus*) from the Sloth Sanctuary, Penshurt, Limón, Costa Rica [6]. The phylogenetic analysis in our study demonstrated that Hepacivirus Cf/NJ12 is independent of the other 14 existing hepacivirus species (Figure 3). However, Hepacivirus Cf/NJ12 formed a large clade with four hepacivirus species *Hepacivirus peromysci* (*E*), *myodae* (*F*), *norvegici* (*H*), *ratti* (*G*), indicating that Hepacivirus Cf/NJ12 may have had a common ancestor with them. Based on the species demarcation criteria of ICTV, the Hepacivirus Cf/NJ12 obtained in this study is a novel hepacivirus species according to p-distance analysis. Notably, these are the first hepaciviruses detected in *Crocidura fuliginosa* and the first whole-genome sequences of the novel hepacivirus reported. The ever-expanding host range of hepacivirus suggests that hepacivirus may have crossed species barriers in ancient times and have thus contributed to its genetic diversification. However, there is no documented evidence of the experimental infection of hepacivirus in laboratory animals or another host other than that in which it was discovered, indicating that the currently occurring hepaciviruses are highly adapted to their specific hosts, and therefore cross-species transmission is less likely.

It is worth noting that rodent hepaciviruses contain six species of hepaciviruses, and bat hepaciviruses contain three species of hepaciviruses, indicating that hepaciviruses are highly genetically diverse in these two animal orders. Furthermore, the phylogenetic relationships revealed that rodent hepacivirus share the earliest common ancestor with other known hepaciviruses [21]. Collectively, rodents and, to a lesser extent, bats may be the ancestor hosts of hepaciviruses [2,15,22]. Nevertheless, our discovery of a new hepacivirus in the insectivores still further deepens our understanding of the genetic evolution of hepaciviruses. Intriguingly, different hepaciviruses can infect the same host. For example, *Hepacivirus ratti* (*G*) and *norvegici* (*H*) infect *Rattus norvegicus* [23]; *Hepacivirus macronycteridis* (*K*) and *vittatae* (*L*) infect *Hipposideros vittatus* [24]. On the one hand, this genetic diversity of hepaciviruses in certain hosts may have evolved through interspecies transmission. On the other hand, it is possible that the segregated animal hosts later come into contact and can still share their divergent hepaviruses.

Nevertheless, there are still many knowledge gaps in the molecular evolution of hepaciviruses, which require further investigation and research [25]. With the focus on the virome in small mammals and the development and advancement of sequencing techniques, it is anticipated that an increasing number of genetically diverse hepaciviruses will be identified in the near future, since hepacivirus has only been detected in less than 1% of the named rodent, chiropteran, and insectivore species to date. Most importantly, the identification of animal hepaciviruses will provide a better understanding of the evolutionary history and origin of human HCV.

We found that in addition to the liver, hepacivirus RNA was also detected in the spleen, kidney, and intestine. Similar to other reported hepaciviruses, the highest viral load was detected in the liver [8,9,15]. However, it should be cautiously interpreted that if hepaciviruses are produced massively in the liver, the viruses are highly likely to enter the blood (viremia), which will lead to RNA detection in the other tissues that have blood. Whether the novel hepacivirus displays hepatotropism remains to be determined using histopathological examinations, immunohistochemistry, and/or in situ hybridization; meanwhile, the impact of hepaciviruses on hosts, such as hepatitis and liver fibrosis, could be studied, which will collectively shed light on virus–host interactions. Unfortunately, due to the improper preservation of tissue samples, we could not perform histopathological examination of the liver tissues of the two hepacivirus-positive samples. The herein-described data were obtained from only one positive animal, which is insufficient for a comprehensive understanding of the epidemiologic characteristics of *C. fuliginosa* hepacivirus.

The discovery of Hepacivirus Cf/NJ12 extends our understanding of the genetic diversity and host range of hepaciviruses. Chimpanzee (*Pan troglodytes*) and tree shrew (*Tupaia telangeri*) are known to have clinical symptoms similar to humans after inoculation with HCV [26,27]. The course and pathological signs experienced by bank vole (*Clethrionomys glareolus*) inoculated with *Hepacivirus myodae* and glareoli *Hepacivirus* are similar to those of humans infected with HCV [28]. However, chimpanzees are relatively rare and expensive subjects, and studying the pathogenesis of HCV replication in chimpanzees may take 10 to 20 years. In fact, the pathogeneses of hepacivirus infection in tree shrews, bank vole, and humans are quite different. Therefore, there is still a lack of suitable animal models to study the pathogenesis and immunity of hepaciviruses. The first discovery of novel hepacivirus in *C. fuliginosa* may provide a new option for future development of a suitable hepacivirus animal model.

## 5. Conclusions

In summary, hepacivirus was identified for the first time in *C. fuliginosa* specimens from Xiangyun County and Nanjian County, Yunnan Province. Hepaciviruses were only identified in *C. fuliginosa* samples, which suggests that more sensitive and powerful detection methods are warranted for an accurate and broader assessment. The obtained genomic sequence of Hepacivirus Cf/NJ12 showed low identity with known hepaciviruses. Thus, it could be classified as a novel species in the genus *Hepacivirus* based on the ICTV criteria [3]. Along with previous reports, we showed that hepacivirus has a wide host range, including *C. fuliginosa* [29,30,31]. Taken together, these results expand our knowledge of the diversity and host range of hepaciviruses. Nonetheless, further studies should address the pathology and evolutionary pattern of hepaciviruses.

## Figures and Tables

**Figure 1 pathogens-12-01400-f001:**
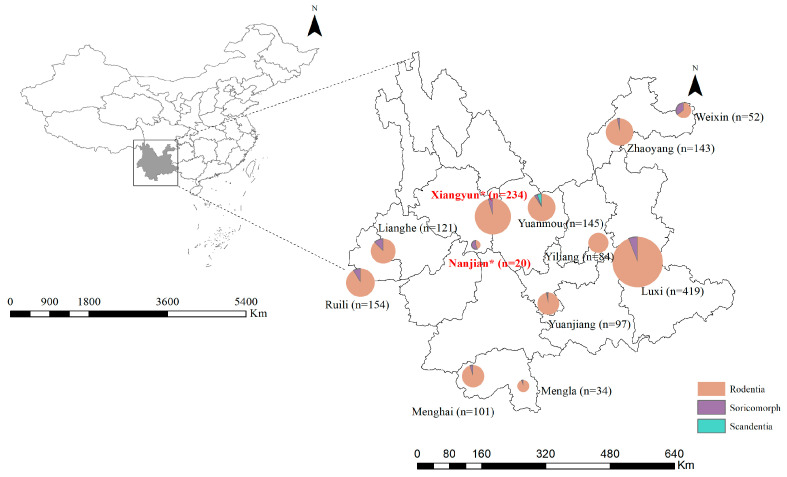
Sampling sites and numbers of sampled specimens per order. The circles indicate 12 sample collection sites in Yunnan Province, China, while red superscript asterisks indicate the spots where the two Southeast Asian shrew hepaciviruses (Cf/XY174 and Cf/NJ12) were detected.

**Figure 2 pathogens-12-01400-f002:**
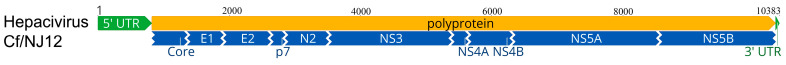
Schematic diagram of the genomic organization of Hepacivirus Cf/NJ12 genome.

**Figure 3 pathogens-12-01400-f003:**
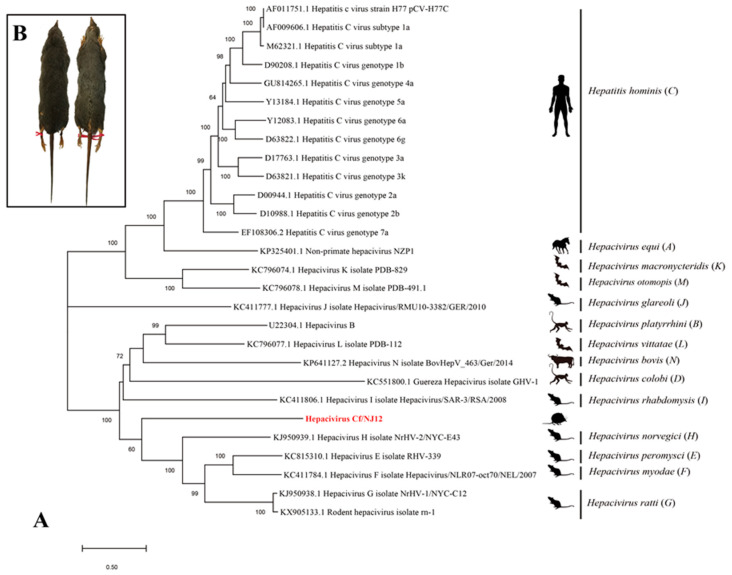
(**A**) Phylogenetic analysis based on the nucleotide sequence of the hepacivirus open reading frame region. Maximum likelihood phylogenetic tree showing the evolutionary relationship of Hepacivirus Cf/NJ12 (highlighted in bold) with representative members of the *Hepacivirus* genus. The scale bar indicates nucleotide substitutions per site. Bootstrap values (≥60) of 1000 replicates are shown in the main nodes. (**B**) Specimen images of the Southeast Asian shrew (*Crocidura fuliginosa*).

**Figure 4 pathogens-12-01400-f004:**
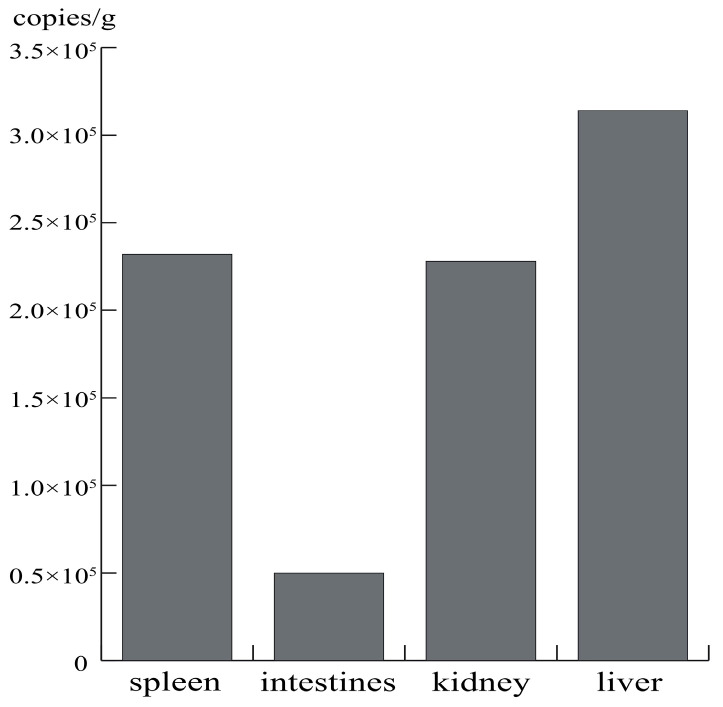
Real-time RT-PCR quantification of viral genomic RNA. The host of positive sample Hepacivirus Cf/NJ12 was selected, and real-time RT-PCR was used to quantify the viral RNA in the liver, spleen, kidney, and intestine. Units are presented in copies per gram.

**Table 1 pathogens-12-01400-t001:** RT-PCR detection results for hepacivirus in small wild mammals collected from 2016 to 2017 in Yunnan Province, China.

Order	Family	Genus	Species	Sampling Location of Samples (Sample Size; Sampling Year)	P/D * (%)
Rodentia	*Cricetidae*	*Eothenomys*	*E. miletus*	Zhaoyang (38; 2016), Luxi (42; 2016), Xiangyun (2; 2016), Xiangyun (7; 2017), Luxi (23; 2017)	0/112 (0%)
	*Muridae*	*Apodemus*	*A. agrarius*	Weixin (3; 2016), Ruili (1; 2017)	0/4 (0%)
			*A. chevrieri*	Lianghe (5; 2016), Yiliang (19; 2016), Zhaoyan (88; 2016), Weixin (1; 2016), Luxi (21; 2016), Xiangyun (4; 2016), Yuanmou (7; 2016), Xiangyun (11; 2017), Luxi (19; 2017), Yuanmou (12; 2017)	0/187 (0%)
			*A. draco*	Menghai (1; 2016), Yuanmou (3; 2016)	0/4 (0%)
		*Mus*	*M. musculus*	Lianghe (12; 2016), Yiliang (2; 2016), Zhaoyang (3; 2016), Weixin (4; 2016), Xiangyun (1; 2016), Mengla (1; 2016), Xiangyun (12; 2017), Yuanjiang (5; 2017), Ruili (17; 2017),	0/57 (0%)
			*M. pahari*	Lianghe (3; 2016), Yiliang (12; 2016), Yuanmou (1; 2016),	0/16 (0%)
			*M. caroli*	Lianghe (1; 2016), Yiliang (26; 2016), Yuanmou (1; 2016), Xiangyun (10; 2017), Yuanjiang (11; 2017)	0/49 (0%)
		*Rattus*	*R. sladeni*	Lianghe (4; 2016), Menghai (3; 2016), Yuanmou (6; 2016), Yuanjiang (11; 2017), Ruili (8; 2017), Nanjian (1; 2017)	0/33 (0%)
			*R. norvegicus*	Yiliang (8; 2016), Zhaoyang (6; 2016), Luxi (202; 2016), Xiangyun (21; 2016), Yuanmou (19; 2016), Xiangyun (32; 2017), Yuanjiang (10; 2017), Luxi (82; 2017), Yuanjiang (11; 2017), Yuanmou (58; 2017)	0/449 (0%)
			*R. rattus*	Lianghe (3; 2016), Ruili (1; 2017)	0/4 (0%)
			*R. tanezumi Temminck*	Lianghe (73; 2016), Yiliang (12; 2016), Zhaoyang (2; 2016), Weixin (19; 2016), Xiangyun (11; 2016), Mengla (26; 2016), Menghai (91; 2016), Yuanmou (10; 2016), Xiangyun (110; 2017), Yuanjiang (42; 2017), Ruili (76; 2017), Luxi (4; 2017), Yuanjiang (3; 2017), Yuanmou (3; 2017), Nanjian (7; 2017)	0/489 (0%)
			*R. nitidus*	Yiliang (1; 2016), Zhaoyang (1; 2016), Weixin (2; 2016), Menghai (1; 2016)	0/5 (0%)
			*R. yunnanensis*	Weixin (5; 2016), Mengla (5; 2016), Yuanmou (7; 2016), Yuanmou (4; 2017)	0/21 (0%)
		*Niviventer*	*N. fulvescens*	Lianghe (1; 2016), Ruili (1; 2017), Nanjian (1; 2017)	0/3 (0%)
			*N. niviventer*	Lianghe (3; 2016), Yiliang (3; 2016), Xiangyun (2; 2017), Ruili (2; 2017), Yuanmou (1; 2017)	0/11 (0%)
		*Bandicota*	*B. indica*	Lianghe (1; 2016), Yuanjiang (1; 2017), Ruili (22; 2017), Ruili (12; 2017)	0/36 (0%)
		*Micromys*	*M. minutus*	Yiliang (1; 2016), Luxi (1; 2016)	0/2 (0%)
Eulipotyphla	*Soricidae*	*Crocidura*	*C. fuliginosa*	Zhaoyang (3; 2016), Weixin (1; 2016), Menghai (1; 2016), Yuanmou (4; 2016), Xiangyun (1; 2017), Yuanjiang (2; 2017), Yuanmou (1; 2017), Nanjian (3; 2017)	2/16 (12.5%)
		*Anourosorex*	*A. squamipes*	Zhaoyang (1; 2016), Weixin (17; 2016), Luxi (12; 2016), Ruili (4; 2017), Luxi (13; 2017), Nanjian (8; 2017)	0/55 (0%)
		*Sorex*	*S. araneus*	Lianghe (15; 2016), Xiangyun (2; 2016), Mengla (2; 2016), Menghai (4; 2016), Xiangyun (7; 2017), Ruili (9; 2017)	0/39 (0%)
Scandentia	*Tupaiidae*	*Tupaia*	*T. belangeri*	Lianghe (1; 2016), Yuanmou (3; 2016), Ruili (1; 2017), Xiangyun (1; 2017), Yuanjiang (1; 2017), Yuanmou (5; 2017)	0/12 (0%)
Total					2/1604 (0.12%)

* P/D is the number of positive samples/sample size.

**Table 2 pathogens-12-01400-t002:** The p-distance values of pp 996–1437 and pp 2693–3115 and sequence similarity in the ORF region of Hepacivirus Cf/NJ12 compared with those of other hepaciviruses.

Virus Strains	GenBank No.	Sequence Similarity (nt/aa)	*p*-Distance Values	
			pp 996–1437	pp 2693–3115
*Hepacivirus equi* (*A*)	KP325401.1	0.326/0.214	0.557	0.568
*Hepacivirus platyrrhini* (*B*)	U22304.1	0.357/0.276	0.549	0.483
*Hepatitis hominis* (*C*)	AF009606.1	0.347/0.206	0.577	0.584
*Hepacivirus colobi* (*D*)	KC551800.1	0.303/0.26	0.534	0.511
*Hepacivirus peromysci* (*E*)	KC815310.1	0.368/0.311	0.543	0.451
*Hepacivirus myodae* (*F*)	KC411784.1	0.377/0.309	0.548	0.439
*Hepacivirus ratti* (*G*)	KJ950938.1	0.367/0.302	0.518	0.464
*Hepacivirus norvegici* (*H*)	KJ950939.1	0.374/0.305	0.534	0.474
*Hepacivirus rhabdomysis* (*I*)	KC411806.1	0.355/0.276	0.573	0.479
*Hepacivirus glareoli* (*J*)	KC411777.1	0.359/0.234	0.590	0.546
*Hepacivirus macronycteridis* (*K*)	KC796074.1	0.341/0.221	0.568	0.566
*Hepacivirus vittatae* (*L*)	KC796077.1	0.365/0.287	0.524	0.508
*Hepacivirus otomopis* (*M*)	KC796078.1	0.335/0.228	0.545	0.580
*Hepacivirus bovis* (*N*)	KP641127.2	0.353/0.254	0.544	0.526

## Data Availability

All sequences used in this study were from the National Center for Biological Information (NCBI) database. Hepacivirus Cf/NJ12 sequences were uploaded to the Genome Warehouse in the National Genomics Data Center under the accession number: GWHBHEC01000000.

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
