# Peer review of "Identification of a Novel Hepacivirus in Southeast Asian Shrew (Crocidura fuliginosa) from Yunnan Province, China"

_pathogens, 2023, doi:10.3390/pathogens12121400_

Round 1
Reviewer 1 Report
Comments and Suggestions for Authors
In the manuscript ‘Identification of a Novel Hepacivirus in Southeast Asian Shrew (Crocidura fuliginosa) from Yunnan Province, China’ authors describe the finding of a novel virus in a new species of shrew. They have performed extensive sampling, and performed valuable work to retrieve the whole genome of one of the two viruses detected, and performed detailed analyses to compared the novel genome to the hepacivirus genomes already known. A qPCR was developed to also quantify virus in different organs of the animals.
I appreciate the work and the novel detection of hepacivirus, which sheds light on genetics of hepaciviruses as a whole. I have several concerns. The initiation of the study is not clear. Why were the small mammals captured and investigated? Will authors also investigate the tissues for other viruses in the future? What are the arguments and ethical considerations to lethally sample such a large number of small mammals. I think with current global biodiversity loss this needs some explanation. The methodology needs a bit more detailed explanation, see more specific comments below, in order to facilitate both the interpretation as well as the repeatability of such studies. If animals were sampled destructively, why was tissue not also sampled in formalin to allow for histopathologic examination, immunohistochemistry and/or in situ hybridization? I think it would make the study more valuable as it might help to see e.g. if liver lesions were associated to the virus presence. One animal (or two) animals already provide so much information of tissue tropism as well as impact for the animal health. The discussion mention several strong statements that in my opinion are not substantiated and need to be reconsidered.
More specifically, please find my suggestions and comments:
Line 24 Shew àShrew
Line 25: ‘To study the prevalence and genetic evolution of Hepacivirus in small 24 mammals in Yunnan Province, China.’ Please revise to make a sentence.
Line 48-49: Not clear what is meant here: ‘Members of different hepacivirus species show characteristically restricted host ranges’ especially with the sentence a few lines later (58-59) ‘study hypothesized that host switches of hepacivirus were possible across host taxa and rodents played a prominent role in the Hepacivirus genealogy’
Line 70: What was the initiation of the study? Were 1602 small mammals caught and euthanized for the detection of hepaciviruses, or was it for answering additional research questions, or pest control? how were they euthanized; why were they only euthanized after transport?
Line 73: Sterile environment is highly unlikely. Aseptic is more likely. What Biosafety level were the carcass sampling performed?
Methods 2.2: Seems some steps are missing. E.g. pre-treatment prior to RNA isolation is missing: how were the tissues processed, weighed on what kind precision scale to get to get to a calculation on the viral copies to gram tissue later. Homogenization, spinning? What was the positive control taken along for RNA isolation as well as the hepacivirus PCR. Eventhough you are looking for new viruses, it is helpful to take a known hepacivirus along with each of the tests to control the PCR worked. Were PCR products ran in gels to see if there were products of appropriate size?
2.3: please indicate if one or two samples were subjected to the described procedures.
Figure 1. Would it be possible to give in the map for each sampling site a circle where the size represents the number of small mammals caught, here you can split the circle in rodentia, shrews and scandentia. I will help to interpret the virus results to geography. The two positive can then be indicated with a star still, providing maybe the two virus names already in the map. Cf/NJ12, Cf/XY174
Table 2: sampling time à sampling year; ‘infection rate’ would leave out, but just provide the % in brackets after the P/D(%). Even if 0%, percentage should be provided for consistency
Figure 3A: please make the virus references a bigger fond for readibility
Discussion:
Line 216-217: .. ‘and their host reservoirs are extensive’; it is not clear what the authors mean with this. Please rephrase to be specific.
Line 217: The list ‘primates, dogs, horses, cows, bats, and rodents’ is wrong in that it compares different taxonomic levels; if you want to start with animal order, all should be animal orders. It leads to the misleading impression among scientists that e.g. ‘bats’ and ‘rodents’ are species, while these groups comprise an extreme variation of mammals and cannot be clustered so easily.
Line 217 and 218: ‘Rodents are the main natural reservoirs of various hepaciviruses;’ Why do authors think this is? Could this have anything to do with the number of mammal species (e.g. Mollentze et al, 2020 – although this is not a zoonotic virus this review suggests that the number of different viruses fit well with the number of species in a particular mammal order) and thus the wide variation in this mammal order? Could the authors maybe speculate on this?
Line 224 &243: shew à shrew
Line 225: ‘It is speculated that the possible reasons are the limitation of sampling sites and sizes’ This needs more explanation and maybe a separate paragraph. Based on hepacivirus prevalences in other species, what is the range of prevalence expected? You have considerable sampling sizes for some species; based on expected virus prevalence in other species, what species you think are underrepresented in your sample set? And what species you think have been sampled in such numbers that your findings suggest there is no hepacivirus occurring – if one would assume the PCR protocol is perfect. In my opinion it deserves mentioning that because the interactions between most of these hepaciviruses and their hosts it is not so easy to understand what small mammals species would likely be able to be a host for this virus.
Line 244: ‘Therefore, it is necessary for long-term surveillance of hepacivirus in their natural reservoirs, and its potential zoonotic transmission risk should be evaluated.’ This is quite a big statement that needs explanation. Do authors think that humans need to survey every virus that occurs in any wild animal species and perform zoonotic risk studies? I do not think this is realistic, nor desirable. To be realistic there should be a focus, maybe based on certain risky virus groups that often switch hosts or have high mutation rates; or on animals that have a high contact rates with humans. To address the biodiversity crises we currently are in, we might argue that it is best to mitigate novel emerging infectious diseases by preventing people to disturb natural areas and if it is necessary to find conscious ways to prevent direct and indirect contacts between people and wildlife (or domestic animals as intermediate hosts). This virus is just another argument to that we should be careful.
Line 264: ‘The ever-expanding host range of hepacivirus suggests that hepacivirus may have crossed species barriers and thus contribute to their genetic diversification.’ Maybe true (although we don’t know how far it goes back to common ancestors), on the other hand these hepaciviruses that are studied a bit, are unable to infect another host than in which they were detected. This suggests that the currently occurring hepaciviruses for which this has been studied (e.g. by trying to infect laboratory animals) have been highly adapted to their hosts and therefore are less likely to cross-species. I think that also deserves mentioning. It might also be important to address how much contacts there are between these shrews and humans in this region to address potential for transmission to people and maybe how authors think this might change in the future due to e.g. climate change. In conclusion I think the factors that make these viruses (un) able to cross species are not known and therefore it is not possible to conclude on its risk.
Line 268: ‘rodents and bats may be the ancestor hosts of hepaciviruses’ could authors explain the underlying reasons/arguments for this statement?
Line 269: ‘However, our discovery of a new hepacivirus in the insectivores suggests that we should intensify our research on hepaciviruses carried by insectivores, which may further deepen our understanding of the genetic evolution of hepaciviruses.’ Why does your discovery in insectivores suggest that insectivores are more likely to carry hepaciviruses? I do not see the relation? Could it be coincidence? And then why would it deepen evolutionary understanding? Would not any new hepacivirus deepen this knowledge? Why only for insectivores?
Line 274: ‘this genetic diversity of hepaciviruses may have evolved genetically through interspecies transmission’ are there also other possibilities for explaining this? Could it also be explained by segregation of populations of the same small mammal species, that later on get back into contact with each other and still being able to share their now divergent hepacivirus strains? Which of the two (or more?) options are more likely?
Line 284: ‘We found that in addition to the liver, hepacivirus RNA was also detected in spleen, kidney and intestine’ If hepaciviruses are produced massively in the liver, the viruses is very likely to get into the blood (viremia), and this will lead to RNA detection in the other tissues that will have blood, this does not suggest a tropism outside the liver! Tropism could be studied by looking with immunohistochemistry and in situ hybridization in formalin fixed paraffin embedded tissues with microcopy. At the same time effect on the host could have been studied (e.g. hepatitis, liver fibrosis) Please add this as a suggestion for further studies into tropism, that can shed light on both virus-host interactions, as well as excretion routes, and hence give us a better understanding on how hepaciviruses interact with hosts.
Comments on the Quality of English LanguageSee my comments in the other part
Author Response
Dear Reviewer:
Thank you for reviewing our manuscript, please see the attachment below, which will be reviewed in conjunction with the 2 word files "pathogens 2664578_Manuscript_Track Changes" and "pathogens2664578_Manuscript_Clean Text".

Reviewer 2 Report
Comments and Suggestions for Authors
Line 246, why the author speculate that this virus may have a potential zoonotic risk in other word is their a reference for this speculation
Was there any pathological or histopathological lesions assessed in these 2 animals?
I am sure the author has uploaded the sequence of the genome of novel virus into the Genbank, but I don’t think that I saw the Genebank # of the sequence mentioned anywhere in the text.
Please add in the M&M how did you assess the presence/absence of recombination, what software did you use?
Comments on the Quality of English LanguageLine 37, long sentence.
editing of English language required
Author Response

(The authors gave the same response as above.)

Round 2
Reviewer 1 Report
Comments and Suggestions for Authors
I thank the authors for replying to my concerns in detail and I am happy with their adjustments and additions. I have just one small comment: deep isoflurane anaeasthesia is not a euthanasia method on it's own. If the animal is bled (exsanguination) while in anaesthesia, this is the euthanasia. So the method would be 'exsanguination during isoflurane aneasthesia'.
Not related to the manuscript but might be valuable to consider for future work, two remarks: 1. although euthanasia was better in the laboratory the amount of stress for rodents to be in captivity and transport (quite a distance also in this study) wouldn't weigh upto it in my opinion. I would be reviewing the ethical approach, I would have wanted the animas to be euthanised on capture site (e.g. cervical dislocation after induction anaesthetics, or induction and then exsanguination) 2. we spike the part of processed tissue in lysis buffer that we will use for RNA extraction with another virus, so that we have an internal control for RNA extraction, and with qPCR even the efficiency of it. We usually run a known positive sample with the other samples tested with the same PCR mix to have a control for the PCR.
Good luck with your future work.